# EU Net-Zero Policy Achievement Assessment in Selected Members through Automated Forecasting Algorithms

Cristiana Tudor [1,*] and Robert Sova [2]

1  International Business and Economics Department, Bucharest University of Economic Studies, 010374 Bucharest, Romania
2  Department of Management Information Systems, The Bucharest University of Economic Studies, 010374 Bucharest, Romania; robert.sova@ase.ro
*  Correspondence: cristiana.tudor@net.ase.ro

**Abstract:** The European Union (EU) has positioned itself as a frontrunner in the worldwide battle against climate change and has set increasingly ambitious pollution mitigation targets for its members. The burden is heavier for the more vulnerable economies in Central and Eastern Europe (CEE), who must juggle meeting strict greenhouse gas emission (GHG) reduction goals, significant fossil-fuel reliance, and pressure to respond to current pandemic concerns that require an increasing share of limited public resources, while facing severe repercussions for non-compliance. Thus, the main goals of this research are: (i) to generate reliable aggregate GHG projections for CEE countries; (ii) to assess whether these economies are on track to meet their binding pollution reduction targets; (iii) to pin-point countries where more in-depth analysis using spatial inventories of GHGs at a finer resolution is further needed to uncover specific areas that should be targeted by additional measures; and (iv) to perform geo-spatial analysis for the most at-risk country, Poland. Seven statistical and machine-learning models are fitted through automated forecasting algorithms to predict the aggregate GHGs in nine CEE countries for the 2019–2050 horizon. Estimations show that CEE countries (except Romania and Bulgaria) will not meet the set pollution reduction targets for 2030 and will unanimously miss the 2050 carbon neutrality target without resorting to carbon credits or offsets. Austria and Slovenia are the least likely to meet the 2030 emissions reduction targets, whereas Poland (in absolute terms) and Slovenia (in relative terms) are the farthest from meeting the EU's 2050 net-zero policy targets. The findings thus stress the need for additional measures that go beyond the status quo, particularly in Poland, Austria, and Slovenia. Geospatial analysis for Poland uncovers that Krakow is the city where pollution is the most concentrated with several air pollutants surpassing EU standards. Short-term projections of PM2.5 levels indicate that the air quality in Krakow will remain below EU and WHO standards, highlighting the urgency of policy interventions. Further geospatial data analysis can provide valuable insights into other geo-locations that require the most additional efforts, thereby, assisting in the achievement of EU climate goals with targeted measures and minimum socio-economic costs. The study concludes that statistical and geo-spatial data, and consequently research based on these data, complement and enhance each other. An integrated framework would consequently support sustainable development through bettering policy and decision-making processes.

**Keywords:** automated forecasting; GHG emissions; European Green Deal; neural network autoregression model (NNAR); statistical methods; aggregated data

## 1. Introduction

Climate change and environmental degradation are currently major concerns for Europe and the rest of the globe [1]. Climate change is considered a serious problem by 93% of Europeans, and almost eight out of ten EU individuals (78%) believe it is an extremely serious issue [2]. In consideration of the above, the European Commission (EC) unveiled

in December 2019 the ambitious European Green Deal [3,4], a package of policy measures spearheaded by the EC with the goal of achieving climate neutrality in the European Union (EU) by 2050 and an intermediate target of cutting carbon emissions by at least 50 percent (and toward 55 percent) relative to 1990 levels by 2030 [5].

Moreover, the pledge toward the green transition has recently been acknowledged as one of the key elements in addressing the economic meltdown caused by the COVID-19 pandemic [6,7] and thus regarded as the lifeline out of the COVID-19 pandemic [1]. Consequently, one-third of the 1.8 trillion Euro investments within the European Recovery Fund announced in July 2020 will finance the European Green Deal's objectives [1,8].

The European Council, comprising the heads of the EU's Member States, further approved the new binding EU objective of at least 55% reduction in GHG emissions by 2030 in December 2020 [8]. Finally, the European Climate Law adopted in June 2021 enforced the European Green Deal objective of making Europe's economy and society climate-neutral by 2050, including its intermediate 2030 goal to reduce net greenhouse gas emissions by at least 55 percent from 1990 levels [9–11].

As such, Europe has long positioned itself as the frontrunner in the global fight against climate change and is making a successful transition to a low-emissions economy [12]. Figure 1 confirms that the EU managed to reduce its greenhouse gas emissions by 23% between 1970 and 2018, whereas the world average registered an increasing trend (i.e., a 69.54% growth) over the same period.

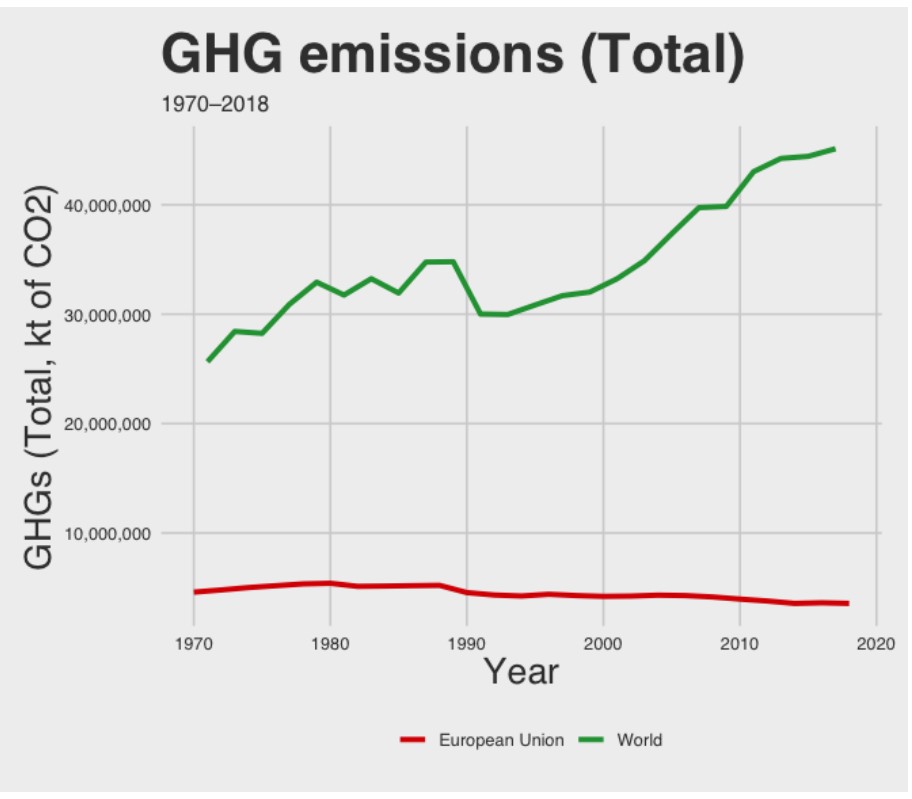

**Figure 1.** Historical evolution of GHGs over 1970–2018. Authors' representation. Data source: World Development Indicators (WDI).

While under the global reach of the Paris Agreement [13,14], individual countries' pledges to reduce their total emissions (i.e., National Determined Contributions, or NDCs) are not legally binding or enforceable [15], the emission reduction targets under the EU Green Deal are mandatory for EU member states [16]. This could, by itself, pose significant problems for Europe's more fragile economies in the CEE area that have not yet been able to catch up with their western counterparts [17–19]. Moreover, the COVID-19 induced

recession is expected to be significantly more severe in smaller CEE economies compared with in the more developed EU states [20], which further adds to the pressure of meeting binding EU climate targets while dealing with the effects of the largest health and economic crisis in recent history [21] and managing post-pandemic recovery.

Although this is a timely issue worldwide, as all countries have to balance post-pandemic recovery and polluting emissions mitigation [22], the CEE economies also deal with the added pressure of legally binding reduction targets. As a result, their implementation may on one hand pose regulatory challenges [23], whereas, on the other hand, noncompliance could have serious repercussions, including infringement procedures, legal challenges, and ultimately financial penalties [24].

Additionally, most governments in the CEE area remain reluctant to implement ambitious pollution reduction policies set by the European Commission, mainly because their economies are still significantly dependent on coal and other fossil fuels. For example, as of 2018, the highest shares of coal in power generation were found in Poland (80%), the Czech Republic (54%), and in Bulgaria (43%) [25].

Poland employs roughly half of the coal workforce at the EU level, followed by Germany, the Czech Republic, Romania, Bulgaria, Greece, and Spain [26], which further complicates the task of implementing the European Green Deal and meeting emission reduction targets. Additionally, citizens in Central and Eastern Europe are generally less concerned about global warming than those in Western Europe [27–30], which also contributes to spurring CEE government reluctance to move toward carbon neutrality.

Although polluting emissions in CEE fell after the fall of the Berlin Wall and the dissolution of the Soviet Union, and the area is currently well behind the world average when it comes to the contribution to world pollution (Figure 2), it should be nonetheless recognized that this is mainly a product of diplomatic compromise and the collapse of an inefficient communist industry, rather than the result of national pollution reduction policies [25], and that the reduction trends remain insufficient compared with the new net 55% reduction target for 2030.

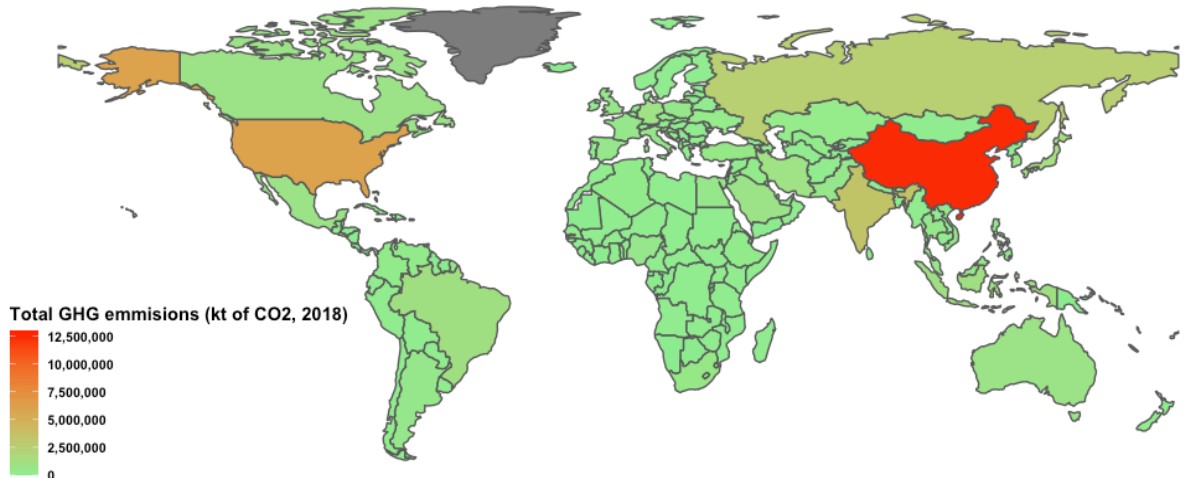

**Figure 2.** Worldwide GHG emissions in 2018. Authors' representation. Source of data: World Development Indicators (WDI).

The formulation of pollution-mitigation policies, as well as its monitoring, intrinsically relies on emissions estimates. Thus, reliable estimates of emission trends are essential for sound policy-making, and national and international bodies are increasingly using forecasts of polluting emissions in their policy-making processes [31]. This, in turn, is a major motivating factor for this research. Furthermore, accurately forecasting emissions and reversing rising trends early-on is especially critical for the more vulnerable EU members, preventing significant costs in the form of infringement procedures, legal battles, and, ultimately, financial penalties. Hence, this study further contributes to assessing

whether the CEE countries are on the right path toward the achievement of their binding pollution mitigation targets, under the assumption that the current conditions and measures remain unchanged.

Consequently, we identify the countries that are farthest from meeting their pollution reduction targets and thus highlight where future research using spatial inventories of GHGs at a finer resolution is needed to uncover specific areas that should be targeted by additional climate combat measures. The results of this research are, on one hand, critical for policy monitoring and its potential revision at the EU level by considering the particularities of the CEE economies and their specific difficulties, and on the other hand for national governments and agencies for early-warning purposes. Overall, we scientifically reason that synergy between EU policy and national measures is key for reaching climate combat goals at the EU level. In turn, GHG emissions forecasts serve as the foundation.

Producing accurate forecasts of polluting emissions remains a challenging research task [32]. Mounting empirical evidence recognizes polluting emissions as a leading factor for socio-economic indicators, including economic growth, mortality, and health variables [33–36]. The univariate time series forecasting allows producing evidence for a leading factor unaffected by other variables and brings further advantages in the form of increased reliability and reduced risk of model misspecification [37].

However, the strand of literature on forecasting polluting emissions over the long term within a univariate framework, particularly with a focus on Central and Eastern Europe, remains thin. This study contributes to filling the void by providing new insights on future emissions trends for a relevant sample of nine CEE countries (i.e., Austria, Bulgaria, Croatia, Czech Republic, Hungary, Poland, Romania, Slovak Republic, and Slovenia).

The univariate methods for forecasting air quality that have been proposed in the extant literature belong to the two major categories delineated by [38], namely statistical methods and deep-learning methods [39]. Ref. [40] used the logistic equation to model $CO_2$ emissions in three Chinese industries and further combined estimations to explain the total $CO_2$ emissions in China.

Ref. [41] employed several econometric methods, including the ARIMA model, Holt–Winters, exponential smoothing, and singular spectrum analysis (SSA). They also introduced a new combination forecast, which includes the SSA and the forecast provided by the Energy Information Administration (EIA). The new combination model was found to out-perform the other candidates and was further used to provide 12-month ahead forecasts for US $CO_2$ emissions energy series.

More recently, [42] used a univariate autoregressive integrated moving average (ARIMA) and several scenarios to forecast $CO_2$ emissions for Pakistan for the 2030 horizon. However, this excludes alternative forecasting methods. Authors that employ machine-learning models generally confirm their suitability for predicting polluting emissions and conclude their forecasting superiority relative to statistical methods.

Ref. [43] used statistical and machine-learning models, including a naive model, ETS, STS, TBATS, HW, ARIMA, and the neural network autoregressive model (NNAR) for modeling and forecasting the evolution of $CO_2$ emissions in Bahrain. They reported that the NNAR model outperformed in the out-of-sample setting model. More recently, [44] modeled and forecast the trend of Bahrain's $CO_2$ emissions by employing multiple forecasting methods (i.e., the Gaussian process regression rational quadratic model, the neural network time series nonlinear autoregressive model, and Holt's method. They also identified the NNAR model as outperforming the other methods by showing the lowest root mean square errors (RMSE) for out-of-sample predictions.

Finally, [45] offered a more extensive study on forecasting polluting emissions. They employed seven statistical and machine-learning methods that are also used in our investigation to predict the GHG emission trends in the twelve world's top polluters. Similar to its predecessors, it also establishes that NNAR is superior in terms of predictive ability at different forecasting horizons by reporting both the lowest root mean square error (RMSE)) and mean absolute scaled error (MASE). Of note, the vast majority of previous research

is usually focused on a single country and stops at exploring the predictive ability of concurrent methods by estimating forecasting accuracy metrics.

This study performs multi-country analysis and further defends its results against the Kolmogorov–Smirnov (KS) predictive accuracy test (KSPA). Additionally, we go further than previous research by monitoring EU net-zero policy implementation within the sample of CEE countries. This is achieved through estimations of GHGs policy-target values that are compared to GHGs estimations produced by the country-specific overperforming forecasting model. A preliminary assessment of the forecasting ability observed for the years 2019 and 2020 is also provided, indicating high accuracy (i.e., 98.5%) in predicting GHG emissions values for the year 2019, albeit, as expected, an overestimation of GHGs values for the year 2020, when the accuracy decreased to 90%, but remained satisfactory, especially in light of the black-swan event.

Furthermore, whereas most related studies employ CO2 emissions as the variable of interest, the current research inputs the more relevant variable of aggregate GHG emissions at a national level. Thus, while acknowledging that estimates of GHG emissions at a national level are less helpful for the development of pollution mitigation strategies and measures [46], we reason that accurate forecasting of aggregate GHG trends still has merits, as it is informative for policymaking purposes.

To this end, six predictive models are implemented through automated forecasting algorithms: the exponential smoothing state-space model (ETS), the Holt–Winters model (HW), the trigonometric ETS state-space model with Box–Cox transformation, ARMA errors, trend, and seasonal components (TBATS) model, the autoregressive integrated moving average (ARIMA) model, and the structural time series (STS) model) within the statistical methods category, and the neural network autoregression model (NNAR) within the machine-learning category.

A naive model, which always predicts the last observed value as per the usual approach in the time series forecasting literature, is also estimated, serving for comparative purposes. The results indicate that the neural network autoregression model (NNAR) shows the best out-of-sample forecasting performance for aggregate GHG emissions at the national level in the sample of CEE countries.

Furthermore, the research findings indicate, based on existing conditions, that CEE countries are projected to miss the mandatory reduction targets under the European Green Deal. Austria and Slovenia are farthest from meeting the 2030's 55% emissions reduction target, whereas Poland (in absolute terms) and Slovenia (in relative terms) are farthest from meeting the EU's 2050 net-zero policy target.

Consequently, the EU's post-COVID recovery plan constitutes a timely occasion to formulate and implement additional measures aimed at putting the CEE countries on the right track toward carbon neutrality. In this respect, this study opens another research direction where the proposed method can be replicated for specific cases on more reliable and finer-scale GHG emission inventories to reveal trouble areas and further intervene with targeted policies and measures, which would minimize social and economic costs for the still fossil-fuel-dependent CEE economies.

For exemplificative purposes, geo-spatial analysis is implemented for the country that is most at risk from meeting EU-set pollution mitigation targets, i.e., Poland. Our findings indicate that pollutants are highly concentrated in and around the city of Krakow, thus, pinpointing where targeted policy intervention should be directed. Forecasts of PM2.5 levels in Krakow indicate an upward trend that is a cause for alarm and highlights that, absent of further policy interventions, the air quality in Krakow will remain under EU standards with negative public health and economic consequences.

For other vulnerable CEE countries, geo-spatial inventories are not publicly available to our knowledge. Nonetheless, among others, [47–49] provided approaches for disaggregating national GHG data that can be further used to perform in-depth analyses for other countries that are projected to significantly miss policy targets under the status-quo hypoth-

esis. This research thus serves to provide a base for future research aimed to complement its findings.

Summing up the above consideration, the main goals of this research are (i) to generate reliable aggregate GHG projections for CEE countries; (ii) to assess whether these economies are on track to meet their binding pollution reduction targets; (iii) to pin-point the countries for which more in-depth analysis using spatial inventories of GHGs at a finer resolution is needed to uncover specific areas that should be targeted by additional measures; and (iv) to perform geo-spatial analysis for the most at-risk country, Poland.

The remainder of the paper is organized as follows. Section 2 explains the data and method employed in the empirical investigation. Further, Section 3 presents the empirical results, performs robustness checks, and provides estimates of emissions trends. Next, Section 4 discusses the main findings, whereas Section 5 concludes the study.

## 2. Materials and Methods

### 2.1. Data

This study uses annual data on aggregate GHG emissions measured in kt of $CO_2$ covering the 31 December 1970–31 December 2018 period, or a total of 49 years, retrieved from the World Development Indicators (WDI) database of the World Bank. The time span for the dataset covers the maximum period of data availability. First, GHG emissions data were extracted from the WDI for all world countries with 49 available annual observations, resulting in a sample of 218 individual time series. Subsequently, data corresponding to nine Central and Eastern European countries that make up the subject of the current research (i.e., Austria, Bulgaria, Croatia, Czech Republic, Hungary, Poland, Romania, Slovak Republic, and Slovenia) was subset. Thus, the final dataset of the study comprises 49 annual observations for nine GHG emissions time series.

The following subsection provides an overview of the state of affairs in the nine CEE countries included in the analysis.

GHG Emissions in Selected Central and Eastern European Countries

Table 1 reflects the total GHG emissions registered during the most recent year of available data, i.e., 2018 in the nine CEE countries, together with the GHG emissions percentage changes over the past decades and policy-relevant periods. The average values at the European Union level as well as at the world level are also reported for comparative purposes.

**Table 1.** Greenhouse gas emissions in nine EE countries: 2018 data and historical evolution.

| Country | Income Category | GHG Emissions 2018 (Total, kt of $CO_2$) | % Change (Relative to 2015) | % Change (Relative to 1990) | % Change (Relative to 1970) |
|---|---|---|---|---|---|
| Austria | High income | 74,980 | −0.66 | −1.68 | 9.6 |
| Bulgaria | Upper middle | 53,330 | −6.94 | −45.77 | −39.42 |
| Croatia | High income | 22,550 | −0.62 | −21.4 | −12.4 |
| Czech Republic | High income | 122,840 | 0.45 | −32.63 | −42.2 |
| Hungary | High income | 60,920 | 4.44 | −31.64 | −30.56 |
| Poland | High income | 389,650 | 6.39 | −11.91 | −16.42 |
| Romania | Upper middle | 109,010 | 0.85 | −55.54 | −38.16 |
| Slovak Republic | High income | 39,930 | 5.72 | −40.39 | −25.37 |
| Slovenia | High income | 17,170 | 5.27 | −3.32 | 54.52 |
| European Union | Aggregates | 3,567,090 | −1.50 | −21.69 | −22.38 |
| World | Aggregates | 45,873,850 | 3.27 | 53.69 | 69.54 |
| | | Average | 1.65 | −27.14 | −15.60 |

Of note, the CEE countries reflected in Table 1 mostly belong to the high-income category as per the World Bank classification, and only Romania and Bulgaria are upper-middle-income economies. As of 2018, the top GHG emitters in the CEE area in absolute terms are Poland, the Czech Republic, and Romania.

Together, the three countries account for almost 70% of 2018 GHG emissions in CEE, with Poland alone responsible for 43.76% of GHG emissions in the area. The vast majority of CEE countries recorded reductions in GHG emissions between 1970 and 2018, with the Czech Republic leading the way (−42.2% between 1970 and 2018), followed by Bulgaria (−39.42%), and Romania (−38.16%). The GHG decrease in these countries is much greater when compared to the 1990 baseline—a development that is directly tied to the demise of inefficient communist industries. Only two CEE countries have shown increases in polluting emissions since 1970, with Slovenia reporting the highest growth rate (54.52%) and Austria reporting a far more modest rate of 9.6%. Over the same time period, at the EU level, GHG emissions decreased on average by 22.38%, whereas the nine CEE countries reported an average reduction of 15.60%.

However, they did manage to surpass the EU average in the aftermath of the communist regime collapse (a decrease of 27.14% for the CEE countries relative to an average decrease of 21.69% at the EU level). Although, over the entire 1970–2018 period, the overall emissions in CEE countries exhibited a declining trend, they did not manage to match their more developed counterparts within the EU.

However, there are disparities in the emission trends at the CEE level, as further reflected in Figure 3, showing that emissions in Poland, Romania, and the Czech Republic remained well above the levels registered in other CEE countries, despite their more accelerated decrease over the past decades.

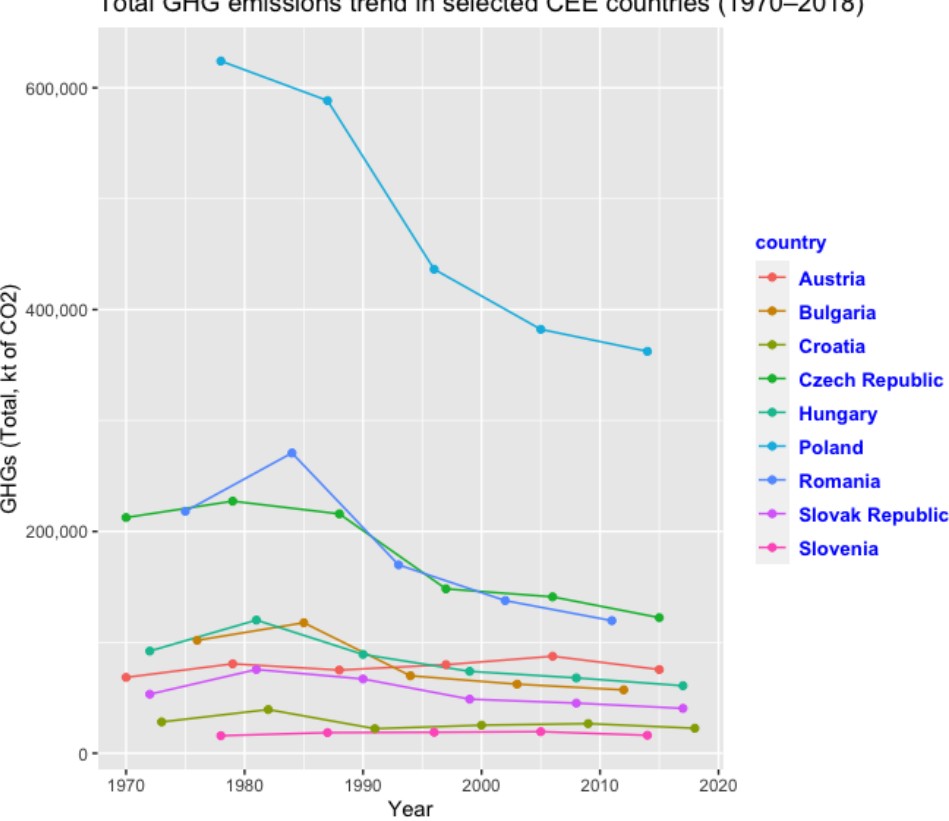

**Figure 3.** Total GHG emission trends in CEE countries. Authors' representation. Source of data: World Development Indicators (WDI).

## 2.2. Method

An overview of the research method implemented to offer an assessment of the EU climate-policy implementation progress in the nine Central and Eastern European countries is first depicted in Figure 4, whereas more details on its main building blocks are provided in the following sub-sections.

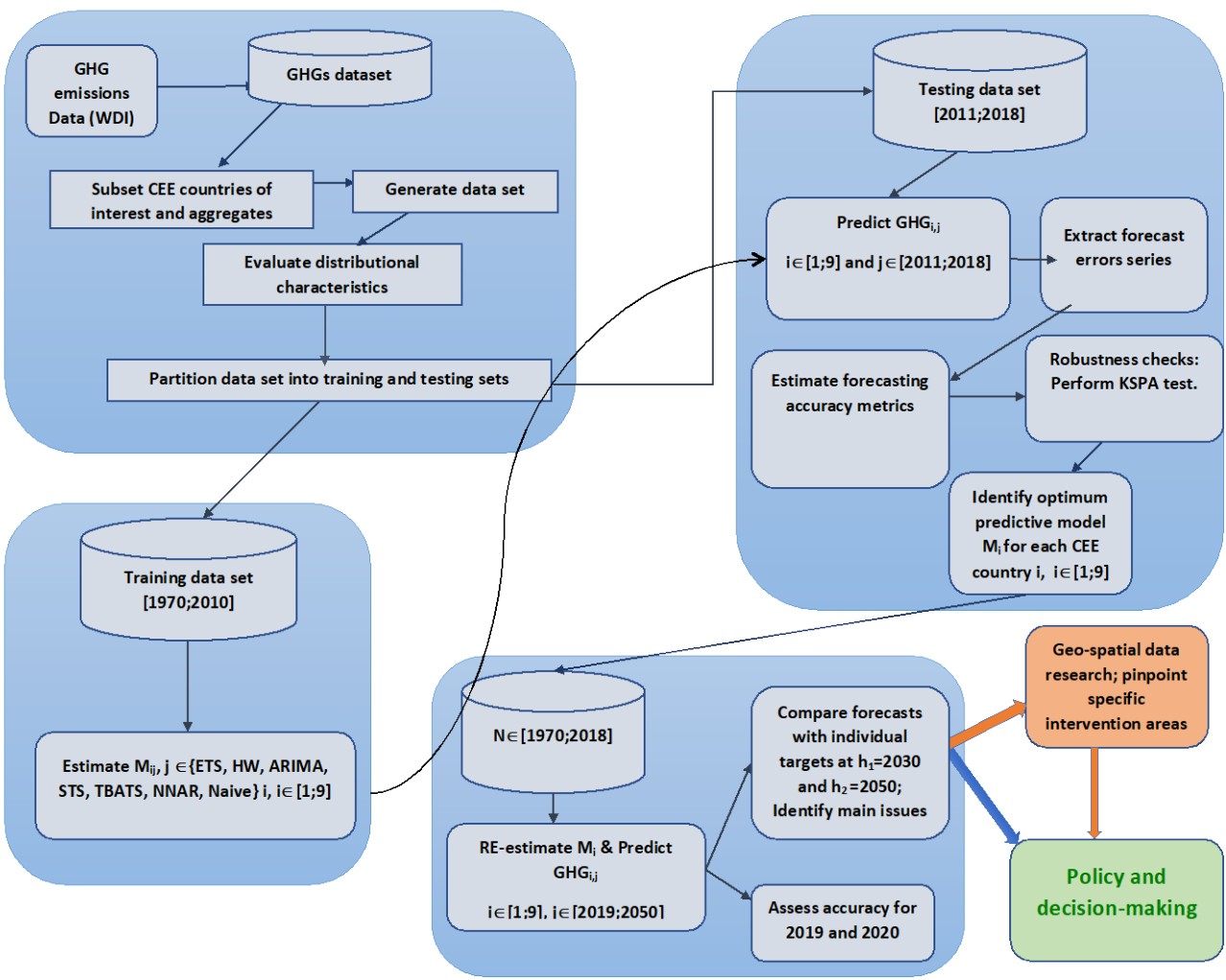

**Figure 4.** Research methodology flowchart.

The research method is based on two crucial elements: (i) the forecasting technique employed to produce forecasts and (ii) the forecasting models encompassed into the algorithm for automated forecasting purposes

### 2.2.1. The Holdout Period Forecasting

For forecasting purposes, the historical data series of length $N_i$ ($i \in \{1, \dots, 9\}$ and N = 49) is separated into two subsets corresponding to a training (or fit) period and a test period or lead-time period to conduct the holdout method [50]. The data up to 2010 (about 84 percent of observations) are employed in-sample for model training and validation, while the data from 2011 to 2018 (roughly 16 percent of observations) are used to assess the predictive models' out-of-sample forecasting accuracy.

As the two most important benchmarks specified within the European Green Deal for emission reduction targets are the years 2030 and 2050, we are particularly interested in finding the best forecasting model within the universe of seven candidates over the lead-time and subsequently using it for providing h steps ahead forecasts corresponding to the two forecasting horizons for GHGs in the nine CEE countries. Thus, *h* is set to 32 (so as

to cover the 2019–2050 forecasting horizon), and the point forecasts for years 2030 and 2050 are highlighted. Figure 5 gives a graphical representation of the holdout cross-validation method [51] employed in this research.

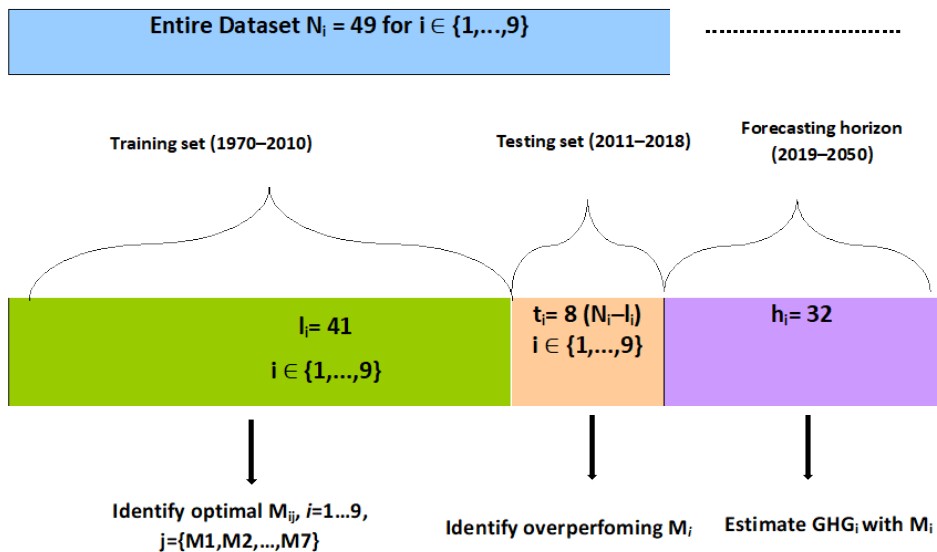

**Figure 5.** The holdout period forecasting method.

### 2.2.2. The Automated Forecasting Models

Machine-learning methods and traditional statistical methods are implemented through automated forecasting algorithms to predict the GHG trends. Figure 6 depicts the univariate forecasting models employed in this study for automated forecasting, delineated in the two categories proposed by [38].

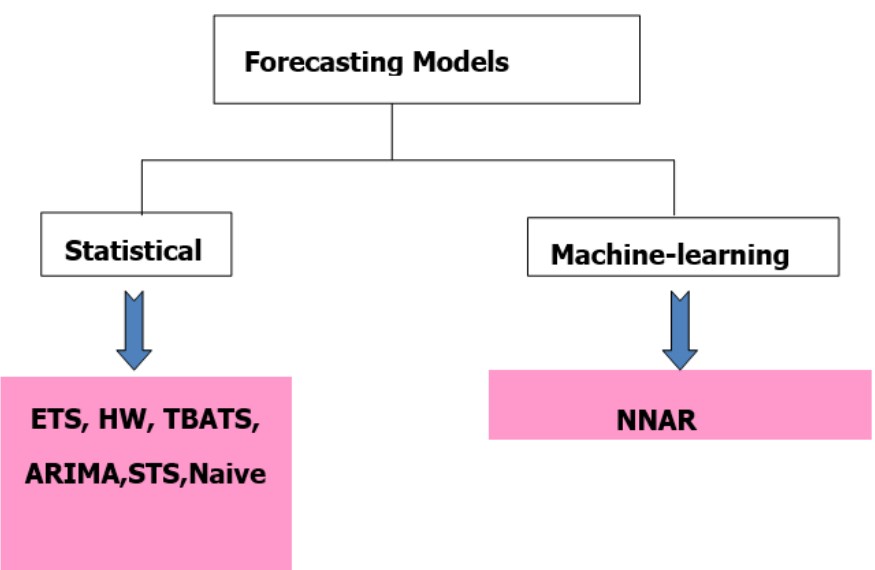

**Figure 6.** The forecasting models employed for automated forecasting.

Hence, the predictive models are automatically implemented in R software through specific functions mainly included in two main packages, the "stats" package [52] and the "forecast" package [53], respectively.

The naive model specified in Equation (1) is manually coded in R such that the k-step-ahead forecast equals the observed GHG emissions value for country $i$ at time $t$, such as

$$F_i(t + k) = y_i(t) \tag{1}$$

A great advantage of the automatization of model estimation is that a specified fitness function (i.e., the AIC, AICc, and BIC) is run to search over a feature space (pool of covariates) to select the optimal model parameters, thus, assuring that the in-sample fit is maximized. Consequently, the algorithm searches through multiple pools of covariates to identify the optimal parameters for 63 models (i.e., 7 models × 9 countries, or $M_{ij}$, i = 1,...,9, j == 1,...,7) and thus identifies parameters of the seven models that assure maximum fit on training data for each country.

The single best-performing model for each country further emerges through the estimation of various accuracy metrics for the out-of-sample predictive ability. Its forecasting superiority is subsequently assessed through formal SPA testing. Further details on implementing traditional statistical methods in the R environment are provided next, while more information can be found in [54,55].

The exponential smoothing state-space (ETS) model was developed by [56] through extending the standard Exponential Smoothing (ES) approach. A forecast equation and a smoothing equation are included in the basic ETS model, which is then incorporated into an innovation state-space model. Each time series is handled as a mixture of three components: the trend (T), seasonal (S), and error (E) components in exponential smoothing, with the trend component further being a combination of a level term (l) and a growth term (b).

The trend and seasonal elements can be none (N), additive (A), additive damped (Ad), multiplicative (M), or multiplicative damped (Md) The final model is a three-character string (Z,Z,Z), with the first letter indicating the state-space model's error assumption, the second indicating the trend type, and the third indicating the season type [53,56,57]). The estimation of the ETS model is entirely automated using R's "forecast" package and its embedded "ets" function. In this research, the system is prompted to choose the error, type, and season autonomously and to identify the optimal parameters by running the corrected Akaike information criterion (AICc) as the fitness function.

*The Holt–Winters Model (HW)* was first proposed by [58,59] and is often referred to as double exponential smoothing. The model smooths the time series with three exponential smoothing formulae that are applied to the mean, trend, and each seasonal sub-series.

The exponential component ($E_t$) is given by

$$E_t = wY_t + (1 - w)(E_{t-1} + T_{t-1}), 0 < w < 1 \tag{2}$$

The trend component is given by

$$T_t = v(E_t - E_{t-1}) + (1 - v)T_{t-1}, 0 < v < 1 \tag{3}$$

Finally, the k-step-ahead forecast is issued as

$$F_{t+k|t} = E_t + kT_t \tag{4}$$

The "HoltWinters" function in the "stats" package in R software is used to automate the estimation of the HW model for the nine annual GHGs time series of interest by performing HW filtering on each series and using the squared prediction error to detect specific parameters.

The TBATS Model (Exponential Smoothing State Space Model with Box–Cox Transformation, ARMA Errors, Trend, and Seasonal Components) was developed by [60]. It can be automatically estimated in R through the "TBATS" function in the "forecast" package. The fitted model is specified as TBATS(*omega, phi, m1, k1 >, . . . , mJ, kJ >*), where *omega* is the Box–Cox parameter; *phi* is the damping parameter; *m1, . . . , mJ* are the seasonal periods; and *k1, . . . , kJ* are the number of Fourier terms utilized for each seasonality. A TBATS model thus requires the estimation of 2(k1 + k2 + . . . kT) initial seasonal values. In our estimations, we instruct the algorithm to employ AIC to automatically identify the model parameters for each of the nine time series (countries).

The ARIMA model (Autoregressive Integrated Moving Average) of [61] is a generalization of the ARMA (autoregressive moving average) model, widely popularized by [62]. A seasonal model is expressed as ARIMA (p,q,d)(P,Q,D)s, where s signifies the seasonal period and the lowercase and capital letters reflect the number of nonseasonal and seasonal parameters for each of its components, as per [54]. An ARIMA(p,d,q)(P,D,Q)s model is expressed in equation form as:

$$(1 - \varphi_1 B - \ldots - \varphi_p B^p)(1 - \Phi_1 B^s - \ldots - \Phi_P B^{sP})(1 - B)^d(1 - B^s)^D Y_t = \\ (1 - \theta_1 B - \ldots - \theta_q B^q)(1 - \Theta_1 B^s - \ldots - \Theta_P B^{sQ})\varepsilon_t \tag{5}$$

where $\varepsilon_t$ is a random variable that has a mean of zero and the standard deviation $\sigma$.

The "auto.arima function" within R's software's "forecast" package is run to return the best ARIMA model for each series, by using unit root tests, AICc minimization, and MLE. The function can determine whether the data that are employed to fit the models require seasonal differencing and estimates unit root tests, while minimizing the AICc and MLE, to select parameters in a step-wise manner. As a result, instead of examining every potential combination of p and q, the "auto.arima" function yields substantially greater efficiency.

Structural Time Series Models (STS), developed by [63] are expressed explicitly in terms of non-obvious components, such as trends, cycles, and seasonals, which have a natural interpretation and represent the key elements of the series under study. The primary concept behind structural time series models is that they are built up as regression models with time-dependent explanatory variables and coefficients that change over time [64]. Equation (6) gives a generalized expression for the decomposition of a time series, such that

$$y_t = \mu_t + \psi_t + \gamma_t + \varepsilon_t, t = 1, \ldots, T \tag{6}$$

where $\mu_t$ is the trend, $\psi$ is the cycle, $\gamma_t$ is the seasonal, and $\varepsilon_t$ is white noise.

Here, as in [65], STS models are automatically implemented in R by maximum likelihood through the function "StructTS" in the "stats" package. Further information on the automatically estimation and forecasting with STS models in R can be retrieved from [55].

### 2.2.3. Neural Network Autoregression Model (NNAR)

Artificial neural networks (ANNs) have been shown to be able to mimic complicated real-world systems whilst effectively allowing for nonlinearities [66] (Pasini, 2015). The nodes, the network architecture reflecting the connections between nodes, and the training algorithm used to determine values of the network parameters for executing a certain task are the basic elements that describe an ANN [67]. Furthermore, an ANN is known as a multi-layered feed-forward network or multilayer perceptron (MLP) when each layer of nodes receives input from the preceding layer.

The models are called feed-forward because there are no feedback connections, and hence the model's forecasts (i.e., outputs) are not subsequently fed back into itself [68]. When working with time series data, lagged values of the time series are frequently employed as inputs to an ANN structure, resulting in neural network autoregression (NNAR) [69]. NNAR models for seasonal data are written as NNAR (p,P,k)m, where m is the seasonal period, p are nonseasonal lagged inputs for the linear AR process, P are seasonal lags for the AR process, and k is the number of hidden layer nodes.

The equation form of the NNAR model is given by:

$$Y = f(H) = f(W * X + B), X = [y(t-1), y(t-2), \ldots, y(t-p)] \tag{7}$$

where Y is the output vector, X is the vector of inputs comprising the observed data's lagged values, H is the vector of nodes in the hidden layer, f is the activation function applied at H that is a transformation of a linear combination of the X (such as a sigmoid/logistic function), W is the weight matrix between X and H, and B is a bias vector [45].

The R forecast package's "nnetar" function, which is a network training function that changes weights and bias values during training, is used to train the feed-forward neural network. Through this algorithm, the nine univariate GHG emissions series are forecast by feed-forward neural networks (FFNN) with a single hidden layer and lagged inputs using the logit activation function to map the input value into the hidden node's value [70], as depicted in Figure 7.

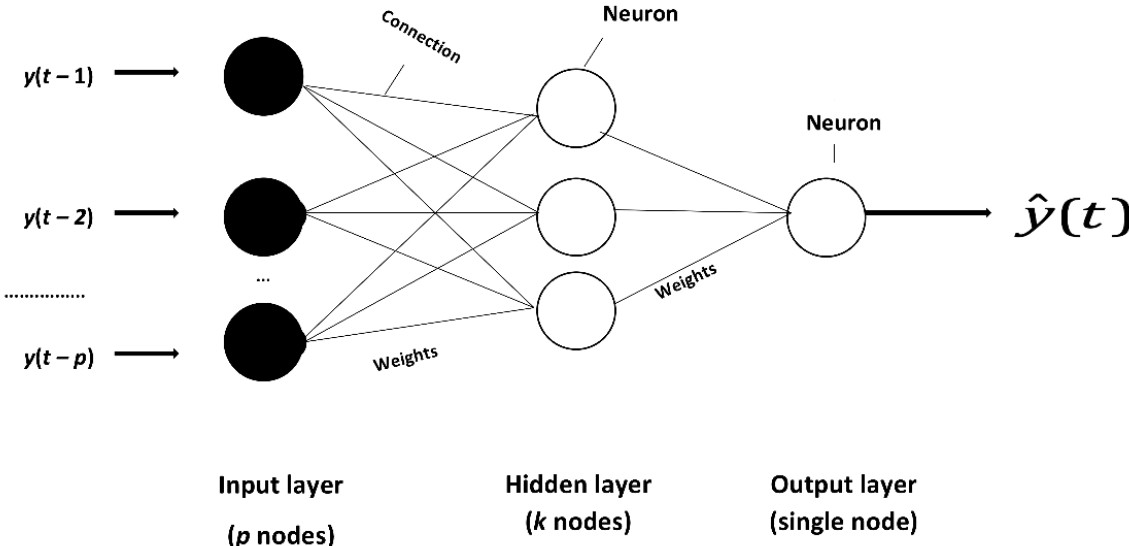

**Figure 7.** A general form of a three-layer FFNN structure with lagged inputs.

Thus, the NNAR model uses *p*-lagged values of the GHG time series as inputs to a neural network with *k*-hidden nodes, for forecasting the output y(t) that contains a single node representing the predicted GHG emissions value. The function automatically makes 25 repetitions (i.e., trains 25 networks with random starting values) and selects the NNAR parameters (i.e., *p* and *P*) through minimizing the AIC, whereas the number of hidden notes is set as $k = (p + P + 1)/2$ (rounded to the nearest integer).

### 2.2.4. Forecasting Accuracy Metrics

The forecasting accuracy of the alternative predictive models for each of the nine GHG emissions series is assessed by estimating the Root Mean Squared Error (RMSE). RMSE is a scale-dependent Goodness-of-Fit (GoF) metric that, due to its benefits, emerged as one of the most popular [71]. Most importantly, RMSE carries the valuable advantage of direct interpretability in terms of measurement units. Although RMSE is less suitable when analyzing multiple time series of different measurement units [72], this is not the case in this research.

Thus, given its suitability and benefits relative to other GoF metrics, RMSE is the forecasting accuracy metric employed in the empirical investigation conducted in this study (similar approaches are found in [37,41,43]. RMSE equals the square root of the mean square error (MSE), and its estimation requires taking the differences between each point forecast and its corresponding value observed over the forecasting horizon, which is further squared and averaged as in Equation (8):

$$RMSE = \sqrt{\frac{1}{N}\sum_{i=1}^{N}(y_i - \hat{y}_i)^2} \tag{8}$$

### 2.2.5. Robustness Checks: The KSPA Predictive Accuracy Test

We scientifically reason that relying on the RMSE alone for identifying the best performing predictive model does not suffice to assure statistical robustness. We thus go one step further and check the statistical significance of point forecasts produced by the over-

performing forecasting model M$_i$, *i* = 1, . . . , 9 by estimating the Kolmogorov–Smirnov (KS) Predictive Accuracy test (KSPA). The KSPA test was developed by [73] as a complement statistical test to the Diebold–Mariano (DM) test [74,75] in order to verify the existence of significant differences between forecasts produced by two competing models.

Thus, whereas the DM approach [76] relies on the mean difference in errors, the KSPA approach is to evaluate the differences in the distribution of forecasting errors. The KSPA test brings some important benefits relative to the popular the Diebold–Mariano (DM) test, including more power, less sensitivity to outliers and increased understanding of the underlying distributional characteristics [73]. The two-sided KSPA test is automatically estimated through the "ks.test" function within R's "stats" package.

## 3. Results

Table 2 reports the RMSE for out-of-sample forecasting results for the test period, or 2011–2018, thus, corresponding to an out-of-sample horizon of h = 8 steps ahead. Within the universe of seven competing models, the NNAR outperforms, showing a superior predictive ability 55% of the time, followed by the HW model (with a 22.22% score). Two statistical models (i.e., ETS and STS) were each found to over-perform in one out of nine instances. The other estimated models (i.e., ARIMA, TBATS, and the Naive model) were not able to produce accurate forecasts for GHG emissions in any of the nine CEE countries considered in this study.

**Table 2.** Prediction accuracy (RMSE for out-of-sample forecasting, forecasting horizon h = 8).

|  | NNAR | ETS | ARIMA | STS | H-W | TBATS | Naive |
|---|---|---|---|---|---|---|---|
| Austria | 2370.88 | 3673.57 | 4139.54 | 4442.13 | 4391.62 | 3492.11 | 4139.54 |
| Bulgaria | 4604.16 | 3847.74 | 4695.38 | 4207.66 | 2824.45 | 4492.23 | 3278.24 |
| Croatia | 2244.37 | 3160.71 | 4794.75 | 3167.93 | 3037.49 | 3105.58 | 3120.77 |
| Czech Republic | 8632.42 | 10,371.08 | 11,512.86 | 4985.65 | 4702.87 | 11,371.10 | 10,371.37 |
| Hungary | 3268.13 | 4345.55 | 9863.11 | 9168.60 | 9051.96 | 7938.42 | 4210.96 |
| Poland | 11,406.20 | 17,777.23 | 23,488.53 | 19,588.15 | 22,488.72 | 28,150.18 | 16,475.80 |
| Romania | 8348.96 | 4934.09 | 7904.91 | 18,473.16 | 18,262.47 | 8938.72 | 6668.08 |
| Slovak Republic | 3588.75 | 4003.57 | 4904.90 | 2301.25 | 2408.13 | 4776.77 | 4003.72 |
| Slovenia | 1856.16 | 1887.70 | 1887.73 | 2750.34 | 2623.70 | 1934.43 | 1888.18 |
| Score * | 5 | 1 | 0 | 1 | 2 | 0 | 0 |
| Score (%) ** | 55.55% | 11.11% | 0% | 11.11% | 22.22% | 0% | 0% |
| Rank | 1 | 3–4 | 5–7 | 3–4 | 2 | 5–7 | 5–7 |

* reflects the number of times (out of nine) that the model surpasses the other candidate models; ** is the percentage of outperformance (within nine iterations, or countries).

Table 3 reports the relative root mean squared error (RRMSE) results for the out-of-sample forecasts corresponding to the nine GHGs time series, where the NNAR model (i.e., the best-performing forecasting model over the lead-time) acts as a benchmark. Hence, the overall performance of NNAR across the nine series was proven to be 29% superior to the ARIMA forecast, 26% superior to TBATS, 15% superior to STS, 10% superior to the naive model, 8% superior to HW, and 7% superior to the ETS model when predicting the trend of GHG emissions in the nine CEE countries.

**Table 3.** The relative root mean squared error (RRMSE) for the forecasting horizon h = 8.

|  | NNAR/ETS | NNAR/ARIMA | NNAR/STS | NNAR/H-W | NNAR/TBATS | NNAR/Naive |
|---|---|---|---|---|---|---|
| Austria | 0.65 | 0.57 | 0.53 | 0.54 | 0.68 | 0.57 |
| Bulgaria | 1.20 | 0.98 | 1.09 | 1.63 | 1.02 | 1.40 |
| Croatia | 0.71 | 0.47 | 0.71 | 0.74 | 0.72 | 0.72 |
| Czech Republic | 0.83 | 0.75 | 1.73 | 1.84 | 0.76 | 0.83 |
| Hungary | 0.75 | 0.33 | 0.36 | 0.36 | 0.41 | 0.78 |
| Poland | 0.64 | 0.49 | 0.58 | 0.51 | 0.41 | 0.69 |
| Romania | 1.69 | 1.06 | 0.45 | 0.46 | 0.93 | 1.25 |
| Slovak Republic | 0.90 | 0.73 | 1.56 | 1.49 | 0.75 | 0.90 |
| Slovenia | 0.98 | 0.98 | 0.67 | 0.71 | 0.96 | 0.98 |
| Average | 0.93 | 0.71 | 0.85 | 0.92 | 0.74 | 0.90 |
| Score * | 7 | 8 | 6 | 6 | 8 | 7 |

* indicates the number of times (out of a maximum of nine) that NNAR outperforms the alternative predictive models.

*Robustness Check: Predictive Accuracy Test*

The Kolmogorov–Smirnov (KS) Predictive Accuracy test (KSPA) proposed by [73] additionally challenges NNAR's over-performance and contributes to ensuring the results' reliability. The KSPA test results are presented in Table 4 for each pair of competing models and each country, with NNAR as acting as a reference. In each instance where NNAR emerged as the top-performing model, the test identified significant disparities between the predictions issued by NNAR and the second-best performing model.

**Table 4.** The results of the Kolmogorov–Smirnov Predictive Accuracy (KSPA) test (*p*-values).

| Country | KSPA (*p*-Value) |
|---|---|
| Austria | 0.01865 *** |
| Bulgaria | 0.1871 |
| Croatia | 0.01865 ** |
| Czech Republic | 0.002486 * |
| Hungary | 0.07937 *** |
| Poland | 0.00235 ** |
| Romania | 0.002486 * |
| Slovak Republic | 0.9801 |
| Slovenia | 0.08702 *** |

Note: * Based on the two-sided KSPA test at a 1% significance level, there is a statistically significant difference between the distribution of forecast errors from the best and second best performing models. ** indicates statistical significance at 5%. *** indicates statistical significance at 10%.

When NNAR did not outperform the best predictive model, the KSPA test was used to determine the differences between NNAR forecasts and the best predictive model. We can reject the null hypothesis and accept the alternate hypothesis when the two-sided KSPA test statistic is significant at 1%, demonstrating that the forecast errors from NNAR and the alternative model do not have the same distribution.

The KSPA test demonstrates that the NNAR forecasting technique outperforms its opponent for all countries (i.e., Austria, Croatia, Hungary, Poland, and Slovenia). The test, on the other hand, does not generally indicate the superiority of the rival model when NNAR is not shown to be the best model in terms of forecasting accuracy, with the exception of the Czech Republic and Romania.

In the last stage of the current research, the over-performing predictive model for each GHG emissions series is fitted to the entire dataset N$i$, $i = 1, \ldots, 9$ and used to generate point estimates for GHGs in the nine countries, underlining the predicted levels for two relevant forecasting horizons, 2030 and 2050, respectively. As such, the entire forecasting horizon is set to h = 32, thus, reaching the 2050 benchmark. The in-sample fit is first confirmed by calculating the Ljung–Box test, which indicates that all models were properly specified (i.e., the residuals were checked for any signs of non-zero correlations at lags 1–20).

Table 5 reflects the percentage change relative to the 1990 level, as specified by the European Green Deal. The results indicate that, on average, the current decreasing trend in total GHGs in eastern European countries will continue over the next decades with a 26.48% decrease until 2030 and an overall 30.02% decrease across the nine countries until 2050. However, the estimation results indicate that two CEE countries (i.e., Austria and Slovenia) will register increases in total GHG emissions of 1.05% and 5.15%, respectively, by 2030 relative to their 1990 levels and similar levels of growth until 2050. Only a couple of countries are expected to reduce harmful emissions significantly and to meet the set pledges for 2030, i.e., Bulgaria (−56.11%) and Romania (−54.80%), although the GHG levels remain too high at the 2050 horizon.

**Table 5.** Forecasts for GHG emissions in nine CEE countries over 2019–2030 and 2019–2050.

| Country | Point Forecast 2030 | % (1990–2030) | Point Forecast 2050 | % (1990–2050) |
|---|---|---|---|---|
| Austria | 77,063.01 | 1.05 | 77,069.90 | 1.06 |
| Bulgaria | 43,162.67 | −56.11 | 26,217.12 | −73.34 |
| Croatia | 26,813.51 | −6.54 | 25,091.39 | −12.54 |
| Czech Republic | 110,686.81 | −39.30 | 90,431.48 | −50.41 |
| Hungary | 58,278.45 | −34.61 | 58,273.77 | −34.61 |
| Poland | 379,509.0 | −14.20 | 379,508.6 | −14.20 |
| Romania | 110,835.6 | −54.80 | 110,968.8 | −54.75 |
| Slovak Republic | 40,879.49 | −38.98 | 42,461.97 | −36.61 |
| Slovenia | 18,674.94 | 5.15 | 18,693.73 | 5.26 |
| Average % change of total GHG emissions | | −26.48% | | −30.02% |

Furthermore, we proceed to explore the preliminary performance of the over-performing models by assessing their forecasting accuracy for the two years subsequent to the dataset spanning period for which estimations of realized GHGs can be realized (i.e., 2019 and 2020). Thus, although official statistics for total GHG emissions are not yet available, data on $CO_2$ emissions for 2019 and 2020 are available for most of the countries included in our data sample.

Thus, in order to provide an estimation for the realized GHG levels in 2019 and 2020, the following approach was implemented: first, the growth rate in $CO_2$ emissions for the years 2019 and 2020 was extracted from $CO_2$ data, and then the same growth rate was applied to the GHG 2018 level. Thus, estimations of total GHGs for 2019 and 2020 were produced by assuming that the total GHGs will register a similar evolution to $CO_2$ emissions. Albeit imperfect, we scientifically reason that this approach is able to provide a sufficiently accurate approximation of the realized GHGs values and can capture the trend particularly well. In a few instances where national reports or other national sources that provide information for actual GHG data were found, the data extracted from those sources were employed.

Table 6 summarizes these findings and indicates all specific sources. The results show high accuracy of the forecasting models in predicting GHG emissions values for the year 2019, albeit, as expected, they overestimated GHG values for the year 2020, when the pandemic was a black-swan event that significantly affected the global GHG levels in 2020. As a result, the confidence level decreased from 98.5% for 2019 to 90% for the year 2020, registering an average value of approximately 95% over the two years. Considering the unseen effects of the COVID-19 pandemic, it can be extracted that the over-performing models are able to provide accurate forecasts for emission trends during "normal" conditions and thus can assist the policymaking process at the EU level.

**Table 6.** Preliminary assessment of the forecasting performance for aggregated GHGs at the national level (2019 and 2020).

| | Actual Value (A) | Forecasted Value (F) | Error (A–F) | \|Error\|/A |
|---|---|---|---|---|
| Austria 2019 | 79,800 [1] | 77,845.6 | 1954.4 | 0.024 |
| Austria 2020 | 73,600 [1] | 76,331.2 | −2731.2 | −0.037 |
| Bulgaria | 57,200 [2] | 55,829.01 | 1371 | 0.024 |
| | 50,724 [3,*] | 57,318 | −6594 | −0.13 |
| Croatia | 22,652 [3,*] | 23,172 | −520 | −0.023 |
| | 21,633 [3,*] | 24,039 | −2406 | −0.111 |
| Czech Republic | 107,194.07 [3,*] | 123,123 | −15,928.9 | −0.149 |
| | 101,036.45 [3,*] | 125,088 | −24,051.5 | −0.238 |
| Hungary | 60,800 [4,*] | 60,344 | 456 | 0.008 |
| | 58,368 [4,*] | 59,402 | −1034 | −0.018 |
| Poland | 394,000 [4,*] | 384,761 | 9239 | 0.023 |
| | 373,680 [5] | 381,685 | −8005 | −0.021 |
| Romania | 105,997.16 [3,*] | 109,402 | −3404.84 | −0.032 |
| | 101,888 [3,*] | 109,715 | −7827 | −0.077 |
| Slovak Republic | 41,900 [6] | 40,909 | 991 | 0.024 |
| | 35,537.7 [4,*] | 39,716 | −4178.3 | −0.118 |
| Slovenia | 16,601 [3,*] | 17,145 | −544 | −0.033 |
| | 14,883 [3,*] | 17,191 | −2308 | −0.155 |
| MAPE (total) | | | | 0.058 (94.2% confidence level) |
| MAPE 2019 | | | | 0.015 (98.5% confidence level) |
| MAPE 2020 | | | | 0.10 (90.0% confidence level) |

* Estimated value from $CO_2$ emissions trends. [1] Source: Umweltbundesamt. [2] Source: https://www.europarl.europa.eu/RegData/etudes/BRIE/2021/689330/EPRS_BRI(2021)689330_EN.pdf (accessed on 11 February 2022). [3] Source: Our World in Data. [4] Source: Statista. [5] Source: http://seo.org.pl/en/w-2020-r-emisja-gazow-cieplarnianych-w-polsce-spadla-do-37368-tys-kt-ekw-co2/ (accessed on 11 February 2022). [6] Source: https://www.europarl.europa.eu/RegData/etudes/BRIE/2021/698767/EPRS_BRI(2021)698767_EN.pdf (accessed on 11 February 2022).

Figure 8 reflects the GHG forecasts over the 2019–2050 period and also highlights the EU Green Deal 2030 individual targets in terms of the total GHG emissions levels to be achieved by 2030. The countries that are farthest from meeting the targets (in absolute terms) are Austria and Poland, whereas the Czech Republic will continue on the path toward pollution mitigation; albeit, the set targets are not projected to be met.

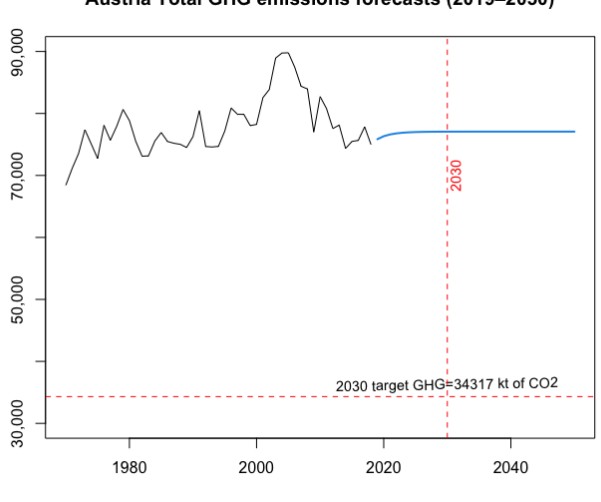

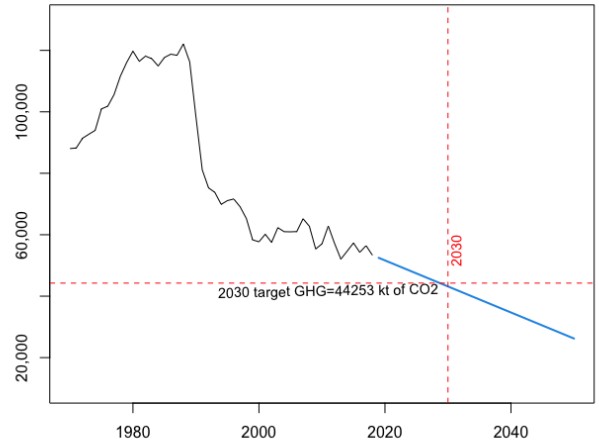

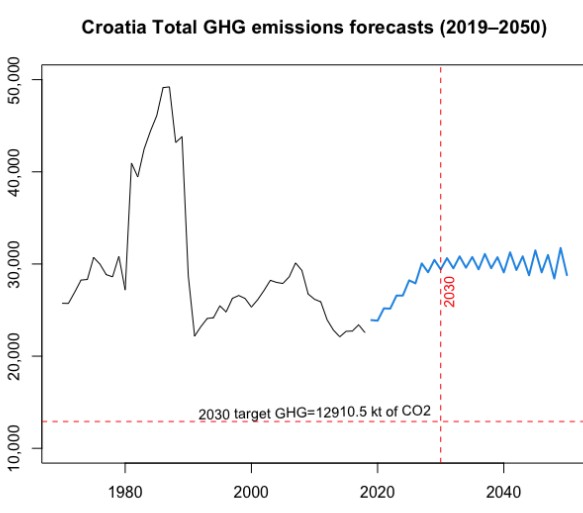

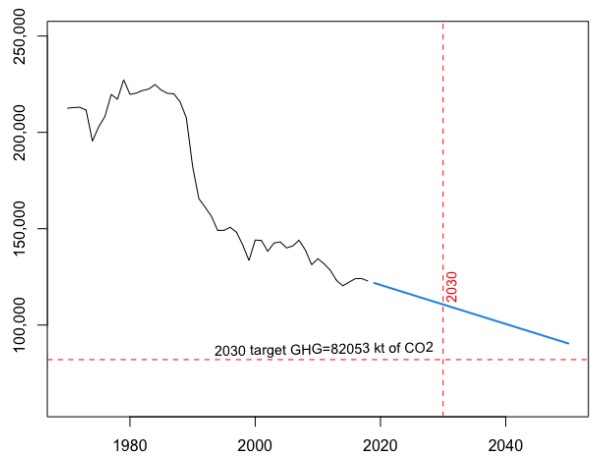

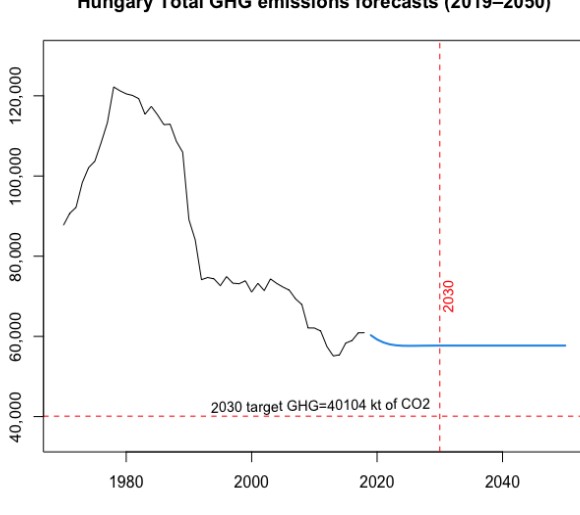

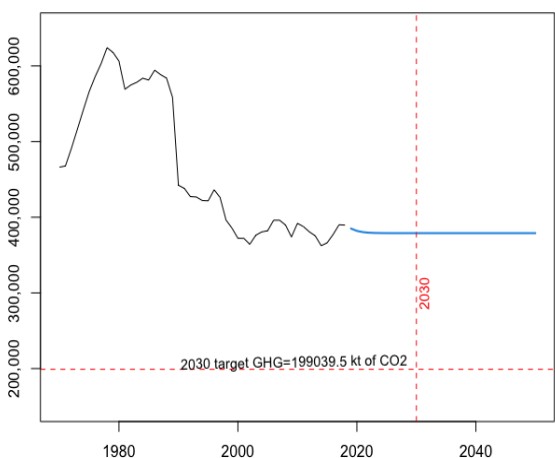

**Figure 8.** *Cont.*

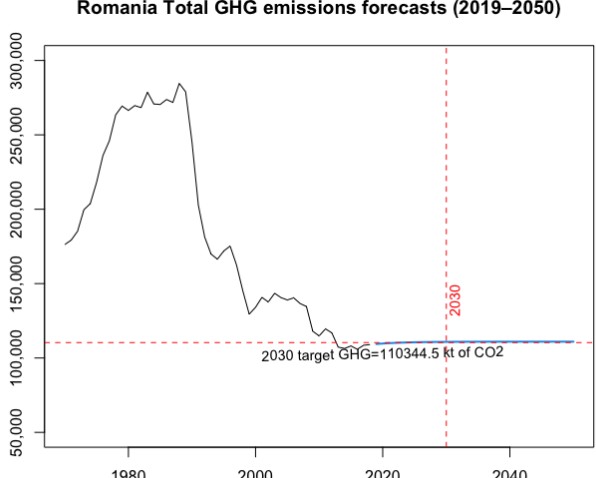

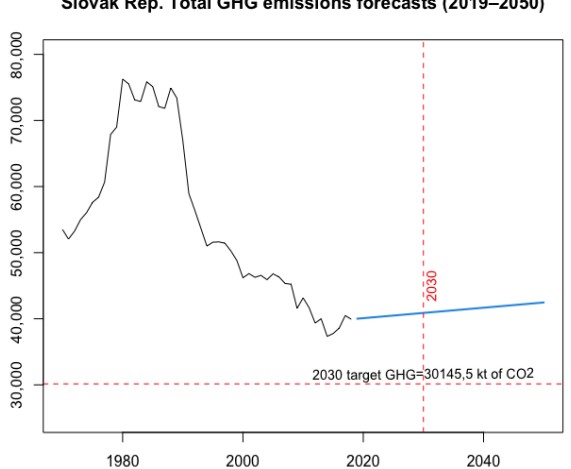

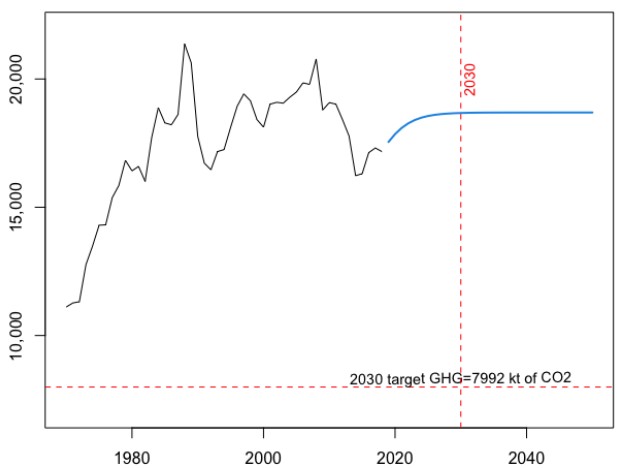

**Figure 8.** GHG forecasts with the best-performing predictive model for each country (actual target values for 2030 are also highlighted). Source: estimation results.

On the other hand, the Slovak Republic is set to reverse its progress toward emissions mitigation, whereas Austria, Croatia, Hungary, Poland, and Slovenia show a future constant evolution in the total GHG emissions. Romania and Bulgaria are the only countries in the CEE area that are expected to meet their 2030 emission mitigation targets.

## 4. Discussion

Our research findings confirm that the neural network autoregressive model (NNAR) exhibited the best out-of-sample predictive ability within the universe of statistical and deep-learning models employed in this study, being able to achieve the lowest forecast error for the nine GHG emissions series 55% of the time. This is in line with [77], who concluded on the efficiency of artificial neural networks (ANN) in time series modeling when a variable's previous values were employed as inputs to describe the future values. Our results also support the conclusion of [43–45], which also identify NNAR as over-performing in forecasting polluting ($CO_2$ and/or GHG) emissions.

Next, we report the total GHG emissions values forecasted with the best predictive model identified for each country. Overall, the estimation results indicate that future emission forecast levels are insufficient to meet the 2050 targets at the CEE level (i.e., carbon neutrality). Our findings thus back the findings of [22] and indicate that the EU climate goals are overly optimistic. A preliminary assessment of the forecasting ability over the

first two time indices in the forecasting horizon (i.e., the years 2019–2020) indicates the high accuracy of the forecasting models in predicting emission values for the year 2019; albeit, as expected, they could not predict the black-swan event that significantly affected the global GHG levels in 2020.

However, it should be mentioned that, to reach carbon neutrality, countries can use carbon credits or offsets from projects that reduce, avoid, or temporarily capture GHGs [78]. Further, net-zero emissions are achieved when all produced GHG emissions are offset by removing GHGs from the atmosphere in a process known as carbon removal. Thus, although carbon neutrality does not imply absolute net-zero emissions, it does, however, require the mitigation of GHGs to as close to zero as possible [79]. Moreover, the most cost-efficient way to address GHG concentrations is by avoiding emissions [80].

One of the main reasons for the previous reduction in GHG emissions within the EU is the increased reliance on renewable energy sources. However, the technical complexities of integrating green sources increase with their share in power generation, and transport, heating, industry, and agriculture are more difficult to decarbonize than the electricity sector, which has been mainly accomplished thus far [81]. Consequently, this might explain why the CEE countries analyzed in this study are projected to miss the ambitious emission targets set through the European Green Deal.

This further confirms that the overwhelming dependency on fossil fuels of CEE economies remains a considerable roadblock to the region's decarbonization goals [82]. In addition, the findings indicate that the 2030 EU-set 55% emissions reduction targets relative to 1990 levels will only be met by Bulgaria and Romania, whereas other countries are expected to continue the decreasing trend without meeting the goal, or even see increasing emissions levels over the next decade.

Our results are thus in line with [83] and confirm that Romania and Bulgaria are committed to mitigating pollution and pursuing low-carbon development [84]. Furthermore, it should be noted that Romania has already met its 2020 renewables target of 24 percent of the final energy consumption from renewables well before the deadline and is also on the right track to reach its 2030 renewables target of 30.7% [85], which helps to explain both the past decreasing GHG emissions trend and its projected evolution.

A similar situation is encountered in Bulgaria, as the country has reached its renewable energy legally binding target for 2020 (16%) since 2013 [86]. Additionally, it was demonstrated that a power system with a substantially higher deployment of renewable energy sources is feasible [87], indicating that more impactful measures can be implemented to accelerate the emissions mitigation trend.

As such, we agree with [88] that an important economic goal is to expedite the transition to renewable energy. This is also in line with previous studies (among others, [89–94] that reported that renewable energy and polluting emissions have a negative association.

Moreover, it should also be mentioned that both Romania and Bulgaria have suffered significant decreases in their population number over the past decades, mostly due to low birth rates and a chronic migration problem (i.e., Romania has the largest negative net migration stocks with the rest of the EU [95]). The demographic decline is also expected to continue until 2050, further explaining the GHG emissions mitigation projected for the countries [96].

However, it should also be acknowledged that, unlike the majority of their former Soviet bloc counterparts, Bulgaria and Romania have not been able to successfully reap the benefits of their industries collapsing with the demise of communism after 1989, which supplied them with millions of tons of excess rights known as Assigned Amount Units (AAUs) [97]. This, in turn, might also explain their more intensive reduction in total GHGs after 1990 relative to other CEE countries.

Nonetheless, the required reduction in greenhouse gas emissions that CEE countries must accomplish comes with important economic costs, which are non-trivial for the more vulnerable EU members [98,99]. Particularly, Poland and the Czech Republic—which are, with Germany, Europe's top three coal-burning countries [100]—will face significant

challenges. Although it can be argued that health benefits can outweigh the costs [101], we take a similar view with [102] and suggest that financial aid and technology transfers are required to support efforts toward carbon neutrality.

The European Union economic recovery package addressing the COVID-19 pandemic's economic and social consequences, i.e., the Next Generation EU (NGEU) fund ([103–105]) should be directed toward implementing targeted climate change and green transition measures and be aimed at achieving a synchronized EU recovery.

To this end, effective measures can emerge from further research based on the current findings and using geo-spatial GHG data for countries that are projected to significantly miss policy targets under the status quo hypothesis. This supports the claims of [106], showing that it is frequently required to collect data not only on the level at which a policy is implemented but also below this level. This would further permit addressing the heterogeneous nature of EU regions and their distinct characteristics and sensitivities to specific policies.

Finally, this research fully agrees with [107] and highlights that increasing ongoing efforts to integrate statistics and geo-spatial data and thus to provide disaggregation at the highest possible level of detail would be conducive to more integrated research endeavors capable of providing reliable and relevant information for effective and efficient decision making, and ultimately support the worldwide goal of sustainable development.

For an exemplification of the aforementioned conclusion, we take a closer look at the country that is most expected to miss the EU-set targets, i.e., Poland. Coincidentally, Poland constitutes a rare case where governmental geospatial data (i.e., a relevant network of air pollution sensors) is publicly available, thus, providing a unique opportunity to identify pollution-related vulnerable areas. Consequently, we sourced pollution data from three government stations located in the three most important Polish urban areas, i.e., Warsaw, Krakow, and Lodz, respectively.

The data includes hourly observations and spans 8 February 2022 to 9 March 2022, for a total of 720 records (i.e., $24 \times 30$) for each of the following air pollutants: PM10, PM2.5, $NO_2$, $C_6H_6$, and CO. The dataset employed for the geo-spatial analysis thus comprises a total of 3600 hourly records. Appendix A Figure A1a–c zooms in at the city level to highlight the exact geo-location of the governmental sensor for which air quality measurement data were extracted. Table 7 centralizes the statistics for all measured parameters by the official government sensors located in the three reference stations over the most recent month.

The statistics in Table 7 demonstrate that the distribution of pollutant data in the three polish urban areas from February to March 2022 was heterogenic, highlighting that Poland's main air quality problem is located in Krakow. The current EU standards that members should meet are, respectively, 40 $\mu g/m^3$ (1-year averaging period) for PM10 concentrations, 20 $\mu g/m^3$ (1-year averaging period) for PM2.5 concentrations, 40 $\mu g/m^3$ (1-year averaging period) for $NO_2$ concentrations, 10 $mg/m^3$ (maximum daily 8-hour mean) for CO, and 5 $\mu g/m^3$ (1-year averaging period) for benzene ($C_6H_6$) [109].

As a result, Krakow is the only area in Poland (among the three sampled here) that currently fails to meet the EU air quality standards for PM10, PM2.5, and $NO_2$, with adverse health effects for its population. This supports [110] and is also in line with a 2016 WHO report (cited by [111]) that positions Krakow as the eighth worst city in the European Union (EU) when it comes to surpassing the thresholds for the air concentration of particulate matter. Figure 9, which offers an overview of pollutant concentration at the country level, further confirms that Poland's pollution is mostly concentrated in and around the city of Krakow.

We next attempt to forecast one of the most dangerous pollutants, i.e., particulate matter 2.5 (PM2.5) for the selected governmental air sensor in Krakow, and we investigate how far the air quality in Krakow will remain from the EU standards over the short term. Particulate matter (PM) refers to suspensions in the air comprising coarse, fine, and ultra-fine solid and liquid particles, whereas PM2.5 refers to PM with a diameter of less than 2.5 $\mu g/m^3$ (i.e., standard air quality measurements describe PM concentrations in terms

of micrograms per cubic meter or μg/m³) that remain suspended in the air for prolonged periods and can cause a variety of short- and long-term health problems. Long-term exposure to PM2.5 has been found to cause, among others, chronic respiratory issues, heart disease, and cancers [39,113]. Overall, it is estimated that particulate matter smaller than 2.5 micrometers (PM2.5) ambient air pollution caused approximately 4.2 million deaths in 2015 ([114,115].

**Table 7.** Geo-spatial data analysis from selected Polish government pollution measurements (8 February 2022–9 March 2022).

| Components | PM10 | PM2.5 | NO$_2$ | C$_6$H$_6$ | CO |
|---|---|---|---|---|---|
| Unit | μg/m³ | μg/m³ | μg/m³ | μg/m³ | mg/m³ |
| Station: Łodz (ul. Gdańska 16) | | | | | |
| Min | 0.5 | NA | 2.2 | 0.3 | 0.2 |
| Max | 136.5 | NA | 82.1 | 10 | 2.4 |
| Mean | 20.7 | NA | 21.8 | 1.6 | 0.5 |
| Station: Warsaw (al. Niepodległości) | | | | | |
| Min | 5.9 | 2.3 | 5.4 | 0.3 | 0.2 |
| Max | 183.2 | 87.4 | 101.3 | 4 | 1.3 |
| Mean | 35.1 | 16.9 | 42.7 | 1.2 | 0.5 |
| Station: Krakow (al. Krasińskiego) | | | | | |
| Min | 3.4 | 2.2 | 6.2 | 0.1 | 0.2 |
| Max | 215.4 | 117.2 | 119.8 | 7.9 | 2 |
| Mean | 52.4 | 27.6 | 51.4 | 1.4 | 0.7 |

Source of data: Poland's Chief Inspectorate for Environmental Protection database (http://powietrze.gios.gov.pl/pjp/home [108] (accessed on 9 March 2022)).

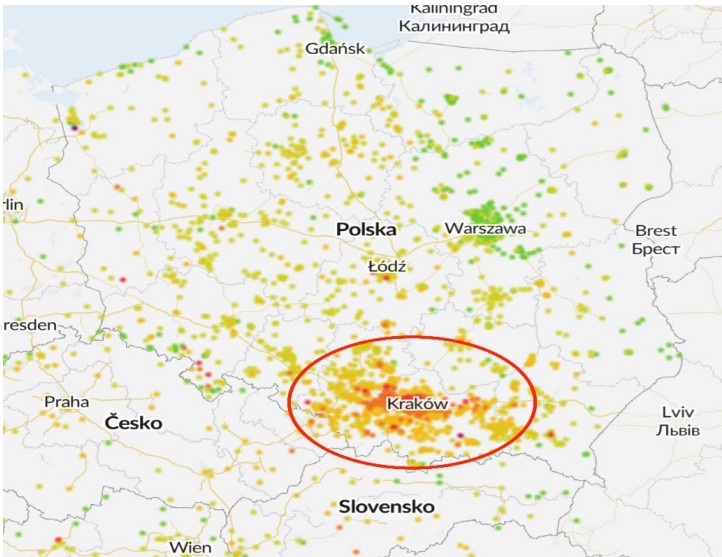

**Figure 9.** Poland air quality concentration map on 9 March 2022. Source: Snapshot of the Map of Air Quality by Airly: https://map.airly.org/ [112] (accessed on 10 March 2022)). Green color: good air quality (low level of pollutants). Yellow color: medium air quality (medium level of pollutants. Orange color: low air quality (high level of pollutants). Red color: very low air quality (very high level of pollutants).

Consequently, we collected hourly PM2.5 levels measured by the government set-up monitoring station over the previous year, thus, constructing a dataset of 8782 hourly observations over the 8 March 2021 (02:00)–8 March 2022 (03:00) period. Figure 10 shows the distribution of the pollutant during this timeframe, reflecting that it remains above the EU-set standards for most of the year, with peaks during the winter months, which in turn indicates that the burning of low-quality coal in coal-fired stoves is an important cause of environmental pollution in the city, in line with [116].

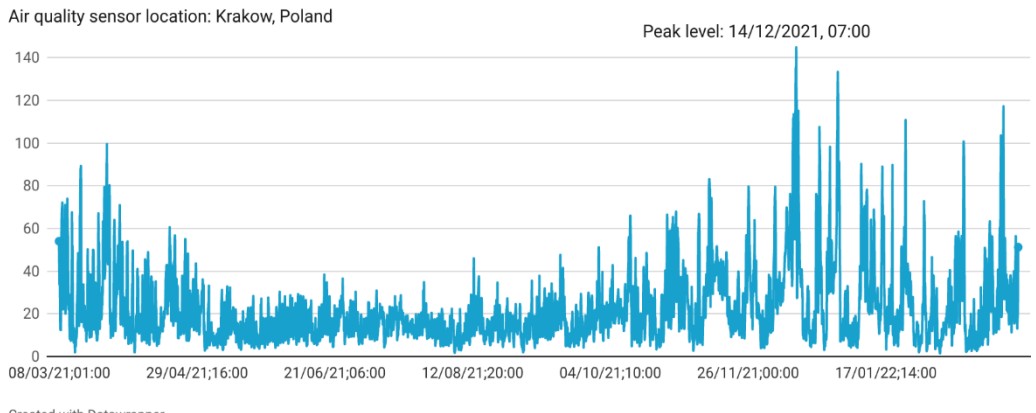

**Figure 10.** The trend of PM2.5 in Krakow during 8 March 2021 (02:00)–8 March 2022 (03:00). Source of data: Poland's Chief Inspectorate For Environmental Protection (http://powietrze.gios.gov.pl/pjp/home (accessed on 9 March 2022)). Authors' representation in Datawrapper.

For estimation purposes, the hourly dataset is subsequently divided into a training set containing the first 7000 observations (i.e., approximately 80% of the data) and a test set of 1782 observations (i.e., 20% of the data) that is employed for assessing the out-of-sample forecasting accuracy of the seven alternative automated forecasting models. Table 8 reports the accuracy metrics for the out-of-sample forecasting of PM2.5, indicating that NNAR is again capable of producing the most accurate out-of-sample forecasts.

**Table 8.** Prediction accuracy metrics (ME and RMSE) for the out-of-sample forecasting * of PM2.5 in Krakow.

| Predictive Model | ME | RMSE |
|---|---|---|
| NNAR | 5.09 | 26.93 |
| TBATS | 13.00 | 28.26 |
| ARIMA | 18.92 | 31.42 |
| Naive | 20.59 | 32.44 |
| STS | 22.08 | 33.40 |
| EXP | −54.12 | 73.95 |
| HW | 94.70 | 106.59 |

* Forecasting horizon of 1782 h (or approximately 74 days); training period of 7000 h (approximately 292 days).

Lastly, the best performing forecasting algorithm (i.e., NNAR) is fitted to the entire dataset containing 8782 hourly measurements for PM2.5 in the Krakow area and instructed to produce forecasted values for PM2.5 over the next 30 days (or 720 hourly observations). Figure 11 reflects the estimation results, along with the benchmark 1-year average level of 20 $\mu g/m^3$ established at the EU level that should be met by all EU members, including Poland.

**Figure 11.** Short-term predictions of PM2.5 in Krakow issued by NNAR(26,14). Source: estimation results. Model Information: Average of 20 networks, each of which is a 26-14-1 network with 393 weight options.

The research results indicate without a doubt that, absent of further policy interventions, the air quality in Krakow will remain low with subsequent negative public health consequences. Thus, forecasts show an upward trend that is a cause for alarm, given that the current quality of PM2.5 levels is already significantly higher than the safe limit of 20 $\mu g/m^3$ prescribed by the EU, which is, in turn, significantly higher than the PM2.5 limit set by the WHO, which recommends a maximum annual mean of 5 $\mu g/m^3$ [113].

Consequently, our research highlights that Polish authorities should take targeted actions to diminish pollution in Krakow. In this respect, we support the conclusion of [111] and agree that, among other actions, informative campaigns and education could make a significant impact on pollution mitigation at the city level. In line with [116], we reason that other measures could focus on restrictions on the use of solid fuel systems, expanding and modernizing heating networks, eliminating waste incineration, and limiting emissions from transportation.

The current research findings further show that failure to implement efficient pollution-reduction actions would result in Poland missing the 2030 and 2050 EU targets with significant negative consequences. Similarly, the availability of granular data for other countries that are also expected to struggle to meet the mandatory pollution-mitigation targets given the status quo (i.e., especially Austria and Slovenia) would prove of great value, as a similar geospatial analysis would pinpoint specific geo-locations where targeted interventions are needed and thus would significantly mitigate the risk of incurring non-compliance-related sanctions.

## 5. Conclusions

The European Union is dedicated to leading the worldwide battle against climate change. Already a lesser contributor to world pollution than the world average, the EU recognized the need to go even further with its climate ambitions to transform Europe into a highly energy-efficient, low-carbon economy. In December 2020, EU leaders agreed to a binding EU goal of a net domestic reduction in greenhouse gas emissions of at least 55 percent by 2030 compared to 1990 and to reach carbon neutrality by 2050. However, reaching these ambitious goals poses significant challenges for the more vulnerable, fossil-fuel-dependent CEE economies, whereas the failure to reach the set pledges might trigger infringement procedures, legal challenges, and ultimately financial penalties.

In this context, accurate predictions of GHG emissions in CEE countries are crucial, which motivated this study. Estimations show that, by 2030, the nine CEE countries would register, on average, a 20% reduction in GHG emissions compared with the 1990 base-year emissions, albeit these reductions remain insufficient compared to the 2030 55% mitigation target.

The only two CEE countries that are expected to meet their emission mitigation goals for 2030 are Romania and Bulgaria, while none of the CEE countries included in the analysis are expected to reach a net-zero target by 2050 by mitigating total GHG emissions to close to zero values in absolute terms. Austria and Slovenia are the farthest from attaining the 2030 objective, while Poland (in absolute terms) and Slovenia (in relative terms) are the least likely to achieve the climate policy aims set for 2050.

Hence, our research findings indicate that achieving carbon neutrality will only be possible through heavy reliance on carbon credits. However, it should be mentioned that forecasting models run under the status quo hypothesis, and thus any future developments are not considered when producing estimates. Consequently, we acknowledge that, given the long forecasting horizon employed in this study, all estimated models should be reviewed and updated regularly to ensure that they remain appropriate for the dynamic circumstances.

Particularly, updating the time series with the inclusion of observations corresponding to the COVID-19 pandemic should lead to more accurate estimations of GHG trends. This study brings evidence based on existing conditions that the current pollution mitigation measures implemented in the selected countries are insufficient, and reaching 2030 targets will require additional efforts that go beyond the status quo.

Nonetheless, a successful EU transition toward carbon neutrality should consider the diverse power systems and starting points in the decarbonization process across CEE member states and should be complemented by adequate social protection programs. Furthermore, as the transition of Central and Eastern European countries to mature and socially just market economies is still underway, the EU's post-pandemic recovery plan should be efficiently used to successfully achieve this transition.

In this respect, this study serves as the foundation for future research aimed at uncovering specific areas that require intervention through research on finer scale geospatial GHG data, particularly for countries projected to significantly miss the EU net-zero policy targets. Currently, this study accomplished monitoring and evaluation of the EU low-carbon policy implementation and identified the requirements for policy response at a national level.

We also performed geospatial analysis for Poland and found that the distribution of the main air pollutants within three major urban areas was highly heterogenic, as a significantly high concentration of pollution was encountered in and around the city of Krakow. Projections of PM2.5 in the city of Krakow showed that the air quality in Krakow will remain too low relative to EU and WHO standards and that immediate pollution-reduction measures are paramount to protect population health and avoid the negative consequences of failing to meet the 2030 and 2050 EU binding targets. For other at-risk CEE countries according to the current findings, further research is needed to fill in the missing pieces once specific geo-spatial data becomes available.

Consequently, we reason that statistical and geo-spatial views complement one another and that additional efforts should be directed toward accelerating their integration. Similarly, research on aggregated statistical data and on (geo) spatial data complement and enhance each other. Thus, the integration of data would, in turn, accommodate integrated research capable of identifying specific issues that require policy responses and thus better assist an effective and efficient decision-making process.

Ultimately, the integrated framework can support the worldwide goal of sustainable development.

**Author Contributions:** Conceptualization, Cristiana Tudor; methodology, Cristiana Tudor and Robert Sova; software, Cristiana Tudor and Robert Sova; validation, Cristiana Tudor; formal analysis, Cristiana Tudor; investigation, Cristiana Tudor; data, Cristiana Tudor; writing—original draft preparation, Cristiana Tudor; writing— review and editing, Cristiana Tudor and Robert Sova; visualization, Cristiana Tudor; supervision, Cristiana Tudor and Robert Sova; project administration, Cristiana Tudor All authors have read and agreed to the published version of the manuscript.

**Funding:** This research received no external funding.

**Institutional Review Board Statement:** Not applicable.

**Informed Consent Statement:** Not applicable.

**Data Availability Statement:** GHG emissions data is available from the World Bank's Development Indicators (WDI) database. Data for air pollutants measured by local sensors in Poland are available from the Poland's Chief Inspectorate For Environmental Protection database.

**Conflicts of Interest:** The authors declare no conflict of interest.

## Appendix A

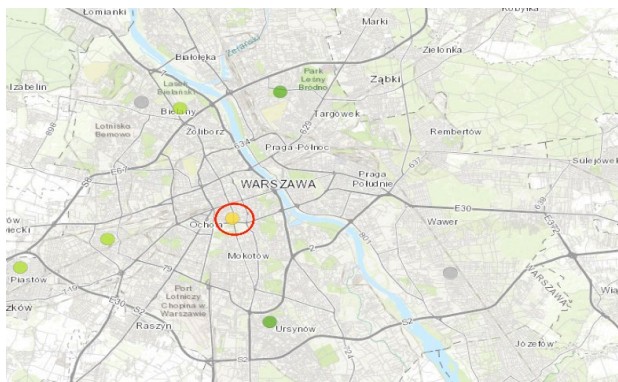

(**a**) Location of the air quality sensor in the Warsaw urban area

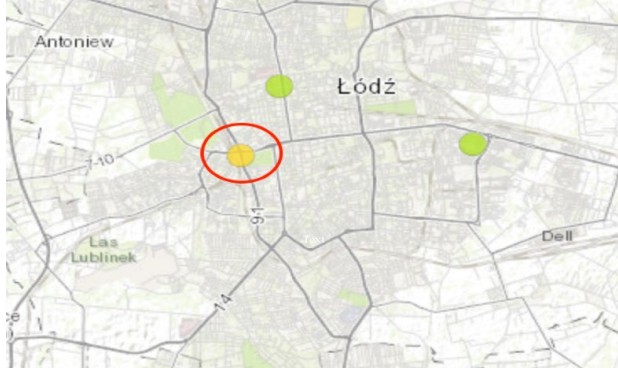

(**b**) Location of the air quality sensor in the Lodz urban area

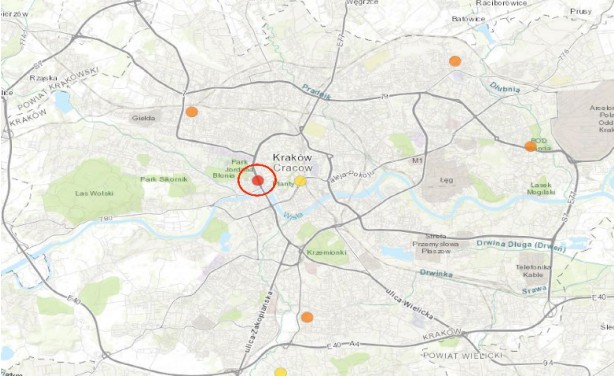

(**c**) Location of the air quality sensor in the Krakow urban area

**Figure A1.** Location of the air quality sensors in Poland's three main urban areas: (**a**) Warsaw, (**b**) Lodz, and (**c**) Krakow. Source: Snapshot from Poland's Chief Inspectorate For Environmental Protection (http://powietrze.gios.gov.pl/pjp/home (accessed on 9 March 2022)). Green color: good air quality (low level of pollutants). Yellow color: medium air quality (medium level of pollutants. Orange color: low air quality (high level of pollutants). Red color: very low air quality (very high level of pollutants).

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
