# Peer review of "EU Net-Zero Policy Achievement Assessment in Selected Members through Automated Forecasting Algorithms"

_ijgi, doi:10.3390/ijgi11040232_

Round 1
Reviewer 1 Report
The paper describes the usage of several predictive models applied to time series of green house gas emissions to make projections until the year 2050 for several European countries.
It seems to be a basic time series analysis using a set of existing algorithms. The paper has a very weak connection to Geographic Information Science (if at all). Hence, I do not think that the paper is suitable for IJGI.
Reviewer 2 Report
Dear authors, Here my comments for improving your work: 1. I think figures 1 and 2 should be updated to 2020 at least 2. minor English corrections are required (i.e. we argue - the meaning should we scientifically reason) 3. CO2 should be CO2 (subscript) 4. Some English sentences are very poor and the meaning is lost.. i.e., pag.5 last sentences and page 6 the first 2... 5. In section 1 the scientific goal of your work should be clearly stated as well as the action taken to achieve it. 6. Please underline the innovative character of your work and what is the contribution for the scholars and academics. 7. The Abstract and Conclusions sections should be improved considering the previous recommendations. 8. Table 1 - why did you choose to present data from 9 EE countries only? And refer just to these later? Is there any particular reason? Maybe you should mention this in your paper. 9. The conclusions section should be split and part of should be included in discussions section. As I already mentioned, this section should be re-written - more concise. Good luck!Author Response
Please see the attachment.

Reviewer 3 Report
The authors in this paper address an important problem of international and urgent concern and aim to generate reliable aggregate GHGs projections for CEE countries. They assess whether these economies are on track to meet their binding pollution reduction targets and pinpoint the countries for which more in-depth analysis using spatial inventories of GHGs at a finer resolution is needed to uncover specific areas that should be targeted by additional measures. They use six predictive models through automated forecasting algorithms namely the exponential smoothing state-space model (ETS), the Holt-Winters model (HW), the trigonometric ETS state-space model with Box-Cox transformation, ARMA errors, trend, and seasonal components (TBATS) model, the autoregressive integrated moving average (ARIMA) model, and the structural time series (STS) model, and the neural network autoregression model (NNAR). A seventh model is a naive model that predicts a flat line and serves for comparative purposes. Their results show that the neural network autoregressive model, NNAR, is overall better at capturing the trends of GHG emissions over the forecasting horizon, with the HW model showing the second-best performance. They claim that the results hold against robustness checks, which allows to confidently employ the over-performing model for each GHG series to issue point forecasts of GHG emissions for the 2050 horizon. All in all, this is a good paper, well-written and presented, with good significance, novelty and contributions. I recommend accepting it subject to the following modifications. Provide more details about the algorithms and methods used in the work as well as the overall methodology. Provide specific details of the methods such as FFNN.
Round 2
Reviewer 1 Report
Thank you very much for this revised version of the manuscript.
As I pointed out in my earlier review, I do not see it as a fit for the ISPRS International Journal of Geo-Information.
The paper investigates the use of several time series forecasting algorithms to make projections regarding green house gas emissions for several European countries.
My main concern is that these forecasting algorithms seem not to take into account the geospatial relationship of those countries, a prerequisite of geospatial analysis.
As geospatial data itself plays a negligible role in this study it thus seems unfit for IJGI and the special issue to which it was submitted.
The newly added discussion on pollution data from Poland only scratches the surface of the use of geodata.
The decision for publication hence primarily lies in the hands of the editors regarding whether they see the article as a fit for the special issue. If they see it a fit, I recommend a minor revision.
However, in any case I would recommend to make some adjustments to the Figures:
Figures 1 and 3: What is the unit of measurement for the y axis? Kilotons? Tons? Please add a label.
Figure 2: Again units of measurement are missing on the legend.
Being a GIS journal, you also should care more about the depicted maps.
What is the map projection? You might want to pick one that preserves areas.
Also, consider normalizing the depicted values (for an explanation, see e.g., here: https://handsondataviz.org/normalize-choropleth.html).
Figure 4 looks sloppy. Please, make sure that all the arrows have a clear start and end point within that Figure.
Figure 6 can probably be condensed to around half its size.
Figure 7 is skewed. Please correct.
Figure 9: What do the colors of the sensors indicate? The legend is missing. Also, they seem somewhat skewed.
Reviewer 3 Report
The authors have addressed my concerns.
Author Response
We really appreciate your insight and acknowledge your constructive criticism.